# Phase and Amplitude-aware Prompting for Enhancing Adversarial Robustness

**Yibo Xu** [1]   **Dawei Zhou** [1]   **Decheng Liu** [1]   **Nannan Wang** [1]

## Abstract

Deep neural networks are found to be vulnerable to adversarial perturbations. The prompt-based defense has been increasingly studied due to its high efficiency. However, existing prompt-based defenses mainly exploited mixed prompt patterns, where critical patterns closely related to object semantics lack sufficient focus. The phase and amplitude spectra have been proven to be highly related to specific semantic patterns and crucial for robustness. To this end, in this paper, we propose a Phase and Amplitude-aware Prompting (PAP) defense. Specifically, we construct phase-level and amplitude-level prompts for each class, and adjust weights for prompting according to the model's robust performance under these prompts during training. During testing, we select prompts for each image using its predicted label to obtain the prompted image, which is inputted to the model to get the final prediction. Experimental results demonstrate the effectiveness of our method.

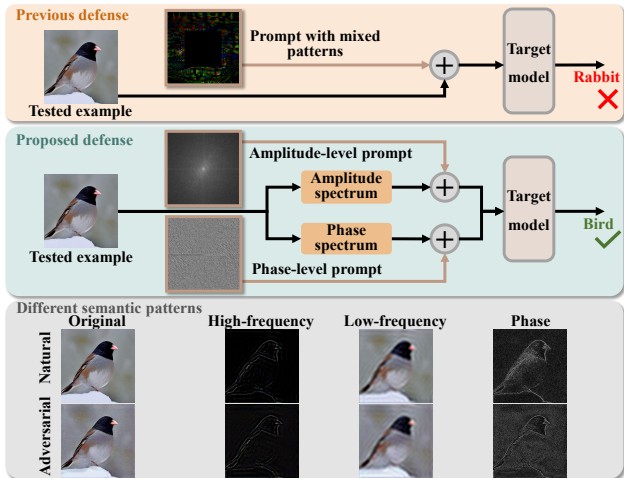

Figure 1. Differences between previous defenses and our defense. Previous method use mixed patterns like pixel or frequency domains for prompting. However, they do not explicitly focus on specific semantic patterns. The phase and amplitude spectra can reflect structures and textures specifically. Our method utilizes these specific patterns for prompting, further improving the robustness.

## 1. Introduction

Deep Neural Networks (DNNs) have been found to be vulnerable to adversarial noises (Szegedy et al., 2014; Xiao et al., 2018; Yang et al., 2023). This vulnerability has posed a significant threat to many deep learning applications (Jaiswal et al., 2022; Mi et al., 2023; Shukla et al., 2024), promoting the development of defenses (Madry et al., 2018; Zhou et al., 2022; Zhao et al., 2024; Xia et al., 2024).

Recently, prompt-based defenses have been increasingly investigated (Huang et al., 2023; Chen et al., 2023; Zhou et al., 2024). It is of interest since it does not retrain target models like adversarial training does (Wu et al., 2020; Wei et al., 2023; Singh et al., 2024), and does not perform major modifications on data as in denoising methods (Nie et al.,

[1]Xidian University, Xi'an, Shaanxi, China. Correspondence to: Nannan Wang <nnwang@xidian.edu.cn>, Dawei Zhou <dwzhou.xidian@gmail.com>.

*Proceedings of the 42nd International Conference on Machine Learning*, Vancouver, Canada. PMLR 267, 2025. Copyright 2025 by the author(s).

2022; Zhou et al., 2023). However, existing prompt-based defenses mainly focus on mixed patterns, such as pixel and frequency domains (see Figure 1). These patterns cannot explicitly reflect specific patterns like structures and textures. To this end, we seek to disentangle the mixed patterns, and construct prompts for stabilizing model predictions by utilizing patterns closely related to the object semantics.

The amplitude and phase spectra of the data have been proven to be able to reflect the specific semantic patterns. Previous studies indicated the amplitude spectrum holds texture patterns (Randen & Husoy, 1999; Sidhu & Raahemifar, 2005), while the phase spectrum reflects structural patterns (Kovesi, 2000; Zhang et al., 2011). Besides, cognitive sciences reveal that people tend to recognize objects by utilizing the phase spectrum (Freeman & Simoncelli, 2011; Gladilin & Eils, 2015), which can also help improve the model's generalization ability (Chen et al., 2021). Also, the amplitude spectrum has been proven to be easily manipulated by noises and thus further processes are needed to mitigate this problem for robustness (Chen et al., 2021). To

this end, *constructing prompts using amplitude and phase spectra is expected to provide positive effects for prompt-based defenses* (see Figure 1).

Motivated by the above studies, we propose a *Phase and Amplitude-aware Prompting* (PAP) defense mechanism, which constructs phase and amplitude-level prompts to stabilize the model's predictions during testing. We learn a phase-level prompt and an amplitude-level prompt for each class, since it can help learn more precise semantic patterns while reducing computational costs compared with learning prompts for each instance. Naturally, a question arises here: *Do amplitude-level prompts and phase-level prompts have the same effect on the model robustness?* To answer it, we utilize phase and amplitude spectra of natural examples to replace the corresponding spectra of adversarial examples respectively for testing as in Table 1. It shows phase and amplitude spectra have different effects on model's predictions. Furthermore, we construct phase-level and amplitude-level prompts, training them under different prompting weights. Table 2 shows different weights lead to different robustness, and thus *we need to adjust their weights appropriately*.

Based on these analyses, we propose a weighting method for our prompts. Since different weights for prompting lead to different robust performances, we adjust their weights based on their influences on robustness during training. We adjust the weight for amplitude-level prompts by the ratio of accuracy under adversarial training examples with amplitude-level prompts to that with phase-level prompts, since the ratio can reflect the relative importance of amplitude-level prompts compared to phase-level prompts.

During testing, we select prompts for each image according to the model's predicted label for it. Previous method (Chen et al., 2023) traverses all the prompts for different classes for testing, causing great time consumptions especially on datasets with many classes. To alleviate this problem, we directly select prompts for tested images according to their predicted labels. To further reduce the negative effect of mismatches between images and selected prompts, we design a loss that helps images with prompts not coming from their ground-truth labels to still be correctly classified. Our code is available at `https://github.com/yeebox/PAP`.

Our contributions can be summarized as follows:

- Considering the amplitude and phase spectra are closely related to specific semantic patterns and crucial for robustness, we seek to design phase-level and amplitude-level prompts to provide positive gains for prompt-based defenses.

- We propose a *Phase and Amplitude-aware Prompting* (PAP) defense. Specifically, we propose a weighting method for prompts based on their impacts on the model's robust performances for training, and propose

to directly select the prompts for images based on their predicted labels for testing.

- We evaluate the effectiveness of our method for both naturally and adversarially pre-trained models against general attacks and adaptive attacks. Experimental results reveal that our method outperforms state-of-the-art methods and achieves superior transferability.

## 2. Related Work

### 2.1. Adversarial Attacks

Adversarial attacks craft malicious noises to mislead target models. White-box attacks like Projected Gradient Descent (PGD) attack (Madry et al., 2018), AutoAttack (AA) (Croce & Hein, 2020), Carlini&Wagner (C&W) (Carlini & Wagner, 2017) and Decoupling Direction and Norm (DDN) (Rony et al., 2019) craft noises through accessing and utilizing models' intrinsic information like structures and parameters. For black-box attacks like transfer-based attacks and query-based attacks (Andriushchenko et al., 2020), attackers have no access to the models' internal information, and thus perform attacks only by interacting with the model's inputs and corresponding outputs.

### 2.2. Adversarial Defenses

Adversarial training methods (ATs) (Madry et al., 2018; Zhang et al., 2019; Wang et al., 2019) aim at augmenting training examples through adversarial noises for training. However, ATs require modifying parameters of models and crafting noises for training, consuming significant resources. In addition, denoising methods (Jin et al., 2019; Zhou et al., 2023) purify images before feeding them into target models. It introduces an additional module for substantially modifying data to remove noises, thereby also consuming great computational resources.

To alleviate this problem, prompt-based defenses has attracted more and more interests due to its efficiency (Huang et al., 2023; Chen et al., 2023; Zhou et al., 2024). C-AVP (Chen et al., 2023) trains pixel-level prompts for each class, and traverses all the prompts for testing. However, it requires high computation costs on datasets with numerous classes. Frequency Prompting (Freq) (Huang et al., 2023) aims at mitigating the vulnerability of models in the high-frequency domain by a masked prompting strategy. However, it does not explicitly focus on specific semantic patterns, where the semantic pattern which C-AVP focused on is also mixed (*i.e.*, the pixel domain). Differently, *we focus on specific textures and structures by prompting on the amplitude and phase spectra*. Also, our method does not traverse all the prompts for testing, *achieving superior performances efficiently through selecting prompts using*

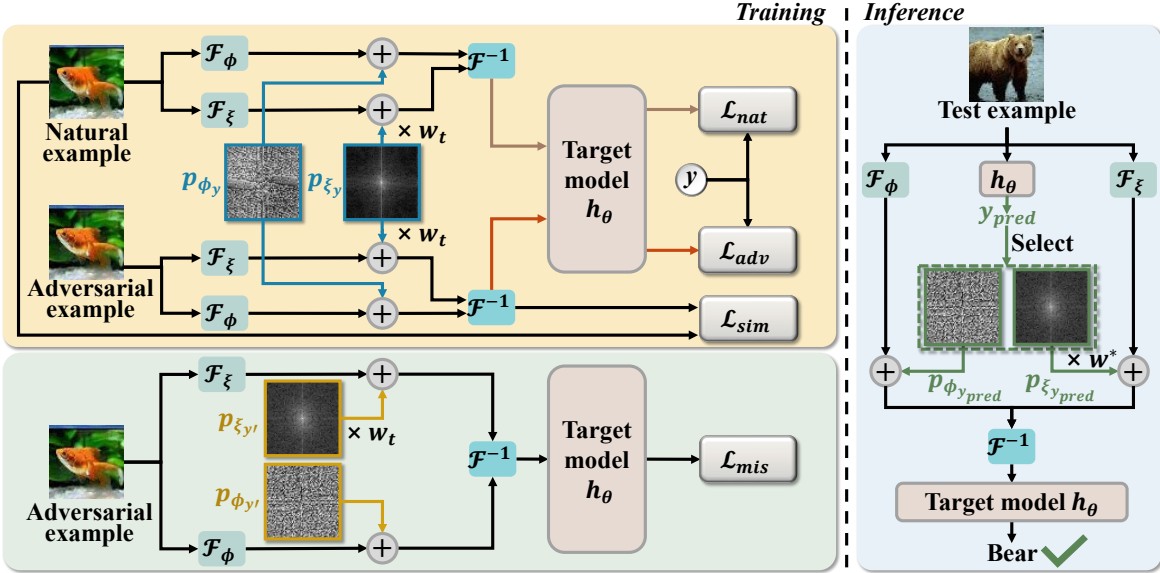

*Figure 2.* The framework of our method. First, We construct phase-level and amplitude-level prompts, and adjust the weights of amplitude-level prompts according to their influences on robustness for training. Then, we select prompts from predicted labels to get prompted images using the finally adjusted weights for testing.

*predicted labels.*

## 3. Methodology

### 3.1. Preliminary

In this paper, we focus on classification tasks under adversarial settings. Given a model $h_\theta$ with parameters $\theta$ and natural data $(x, y)$, the adversarial example $\tilde{x}$ is crafted for misleading $h_\theta$. Since we focus on images, we utilize Discrete Fourier Transform (DFT) and its inverse version (IDFT), denoted as $\mathcal{F}(\cdot)$ and $\mathcal{F}^{-1}(\cdot, \cdot)$, respectively. The phase and amplitude spectra are derived as $\phi_x = \mathcal{F}_\phi(x)$ and $\xi_x = \mathcal{F}_\xi(x)$. We use $\phi_x$ and $\xi_x$ to denote phase and amplitude spectra of a natural image $x$, while $\phi_{\tilde{x}}$ and $\xi_{\tilde{x}}$ denote the corresponding spectra of $\tilde{x}$. In addition, the process to recover an image from its phase and amplitude spectra is expressed as $x = \mathcal{F}^{-1}(\phi_x, \xi_x)$. *Our goal is to design prompts to assist $h_\theta$ in making accurate predictions without the need of the model retraining.*

### 3.2. Motivation

DNNs can be easily fooled by adversarial noises. Defenses like adversarial training and denoising methods all improve robustness with a high computational cost, promoting the development of prompt-based defenses due to the efficiency. However, existing prompt-based defenses focus on mixed patterns like pixel or frequency information, which cannot capture specific patterns like structures and textures (Ying

et al., 2001; Ren et al., 2015). Thus, we seek to further disentangle these semantic patterns for enhancing robustness.

The phase and amplitude spectra have been proven to be able to reflect specific semantic patterns. Through Fourier transform, image signals in the pixel domain can be converted into the frequency domain, which can be further decoupled into phase and amplitude spectra. The phase spectrum can reflect structures (Kovesi, 2000; Zhang et al., 2011), while the amplitude spectrum carries textures (Randen & Husoy, 1999; Sidhu & Raahemifar, 2005). Cognitive sciences indicate people tend to recognize objects by leveraging structures from the phase spectrum (Freeman & Simoncelli, 2011; Gladilin & Eils, 2015), which has been proven to be able to help DNNs improve their generalization performances (Chen et al., 2021). Also, the amplitude spectrum has been analyzed to be easily manipulated by noises, indicating the necessity to mitigate this problem for robustness (Chen et al., 2021). *To this end, constructing phase-level and amplitude-level prompts to disentangle the mixed patterns is considered to be beneficial for improving prompt-based defenses* (see Figure 1).

### 3.3. Defense

Based on the above analyses, we introduce the designed *Phase and Amplitude-aware Prompting* (PAP) defense. We first construct prompts and adjust the weights of amplitude-level prompts based on their influences on robustness. During testing, we select prompts from predicted labels for

prompting. The framework is shown in Figure 2.

### 3.3.1. PROMPT CONSTRUCTION AND TRAINING

We firstly construct and train a phase-level prompt and an amplitude-level prompt for each class, since it can help learn precise natural semantic patterns for each class while reducing computational costs compared with learning prompts for each instance. We randomly sample a natural example $x$ from class $y$, and obtain its phase spectrum $\phi_x = \mathcal{F}_\phi(x)$ and amplitude spectrum $\xi_x = \mathcal{F}_\xi(x)$ as the prompt initialization for class $y$. The prompt initialization for other classes is performed following the above operation. Then, the initialized phase-level and amplitude-level prompts are denoted as $\{p_{\phi_i}\}_{i=0}^{c-1}$ and $\{p_{\xi_i}\}_{i=0}^{c-1}$, where $c$ is the number of classes. The prompted image $x^p$ for $x$ is obtained as follows:

$$x^p = \mathcal{F}^{-1}(\phi_x + p_{\phi_y}, \xi_x + p_{\xi_y}), \quad (1)$$

where $p_{\phi_y}$ and $p_{\xi_y}$ denote the phase-level prompt and the amplitude-level prompt corresponding to the ground-truth label $y$ of $x$. We perform prompting for the adversarial example $\tilde{x}$ following the same way as Equation 1. Then, the designed training losses are introduced as follows:

**Classification Loss.** To enforce our prompts to stabilize the model's predictions, we promote our prompts to learn to help correct wrong predictions of models, and thus exploit the prompted examples to construct the classification loss:

$$\mathcal{L}_{adv} = -\frac{1}{N}\sum_{i=1}^{N}[y_i log(h_\theta(\tilde{x}_i^p))], \quad (2)$$

where $N$ is the number of examples, and $\tilde{x}_i^p$ denotes the prompted image of the adversarial example $\tilde{x}_i$ using the prompts from its ground-truth label $y_i$. Then, the classification loss for the natural prompted data is presented as:

$$\mathcal{L}_{nat} = -\frac{1}{N}\sum_{i=1}^{N}[y_i log(h_\theta(x_i^p))], \quad (3)$$

where $x_i^p$ denotes the prompted image of the natural example $x_i$ using the prompts from its ground-truth label $y_i$.

**Reconstruction Loss.** The constructed prompt could modify the phase and amplitude spectra during prompting. To ensure these modifications do not severely disrupt the original semantic patterns, we design a reconstruction loss between the prompted adversarial images and natural images as:

$$\mathcal{L}_{sim} = \frac{1}{N \times H \times W}\sum_{i=1}^{N}\sum_{j=1}^{H}\sum_{k=1}^{W}e^{|\tilde{m}_{j,k}^p - m_{j,k}|}, \quad (4)$$

where $\tilde{m}_{j,k}^p$ and $m_{j,k}$ denote the pixel value of $\tilde{x}_i^p$ and the pixel value of $x_i$ in the j-th row and k-th colum respectively, and $H, W$ denote the height and weight of the image.

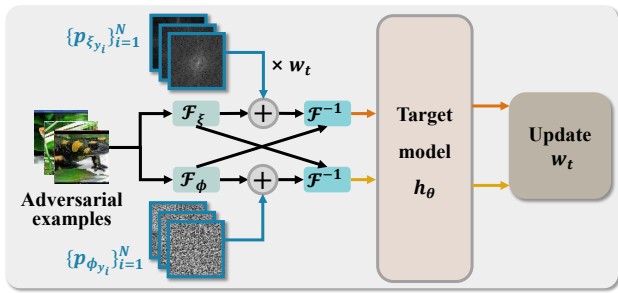

*Figure 3.* The weighting method for amplitude-level prompts. We use amplitude-level and phase-level prompts respectively for prompting, and adjust weights by the ratio of accuracy of images with amplitude-level prompts to that with phase-level prompts.

**Data-prompt Mismatching Loss.** Note that we learn a phase-level prompt and an amplitude-level prompt for each class. Also, we select prompts according to predicted labels during testing. Therefore, there exist mismatches between test images and selected prompts when testing. Previous studies (Chen et al., 2023) indicate we can get prompted images using prompts from classes different from ground-truth labels, and enforce their outputs on ground-truth labels to be larger than those on other labels, so that these images can still be correctly classified to some extent. To this end, we construct a data-prompt mismatching loss as:

$$\mathcal{L}_{mis} = \frac{1}{N}\sum_{i=1}^{N}\{max\{h_\theta^{y_i'}(\tilde{x}_i^{p'}) - h_\theta^{y_i}(\tilde{x}_i^{p'}), -\tau\}\}, \quad (5)$$

where $\tilde{x}_i^{p'}$ is the prompted adversarial example using prompts from $y_i'$, which is a randomly selected label different from its ground-truth label $y_i$. $h_\theta^{y_i'}(\cdot)$ and $h_\theta^{y_i}(\cdot)$ denote outputs on $y_i'$ and $y_i$, and $\tau$ is a threshold.

### 3.3.2. WEIGHTING METHOD

The designed prompts with the corresponding training procedure focus on helping the model make predictions accurately during testing. However, it is natural for us to question *whether the amplitude-level and phase-level prompts have the same influence on the robustness*. To this end, we conduct several experimental analyses to answer it.

We firstly replace the amplitude and phase spectra of adversarial examples with the corresponding spectrum of natural examples respectively. As shown in Table 1, for different models, the natural amplitude spectrum and natural phase spectrum contribute differently to the model's robust performances. It indicates that *the amplitude and phase spectra have different influences on the model's robustness*.

Due to their different influences on robustness, we may need to assign different weights for phase-level prompts and

*Table 1.* The impact of different spectra on robustness. *Adv. All* denotes normal noises. *Nat. Pha./Amp.* indicates we replace phase/amplitude spectra with corresponding natural spectra.

|       | Adv. All | Nat. Pha. | Nat. Amp. |
|-------|----------|-----------|-----------|
| NAT   | 0.00     | 47.81     | 13.41     |
| AT    | 46.82    | 69.00     | 70.10     |

*Table 2.* Robust accuracy (percentage) under different weights. $\alpha$ and $\beta$ are weights of phase-level and amplitude-level prompts.

| $\alpha$ | $\beta$ | NAT | | AT | |
|------|------|-------|-------|-------|-------|
|      |      | None  | PGD   | None  | PGD   |
| 1    | 0.01 | 88.82 | 34.44 | 84.31 | 47.69 |
| 0.01 | 1    | 91.11 | 5.64  | 73.84 | 50.87 |

amplitude-level prompts to further improve the robustness. To show this, we assign different weights for them to obtain the prompted images for training and testing. As shown in Table 2, it is clear that different weight assignments result in different robust performances. Therefore, *we need to design a strategy which can appropriately adjust their weights.*

Based on it, we enforce prompts to assign weights for themselves according to their influences on the robustness. The robust accuracy after prompting can explicitly reflect the influence of these prompts on robustness, which has been proven to be suitable for measuring the importance of them and adjusting their weights (Wang et al., 2019; Wei et al., 2023). *Therefore, we use the robust accuracy under these prompts to adjust their weights.* Specifically, during training, we obtain robust accuracies in training data using amplitude-level and phase-level prompts for prompting respectively. Then, the weights for amplitude-level prompts are adjusted by the ratio of accuracy under amplitude-level prompts to accuracy under phase-level prompts, since it can reflect the relative importance of amplitude-level prompts compared with phase-level prompts for robustness. The weight strategy is specified as:

$$w_t = w_{t-1} \times \frac{\sum_{i=1}^{N} \mathbb{I}(f(\tilde{x}_i^{p_\xi}) = y_i)}{\sum_{i=1}^{N} \mathbb{I}(f(\tilde{x}_i^{p_\phi}) = y_i)}, \quad (6)$$

where $\tilde{x}_i^{p_\xi} = \mathcal{F}^{-1}(\phi_{\tilde{x}_i}, \xi_{\tilde{x}_i} + w_{t-1} p_{\xi_{y_i}})$ and $\tilde{x}_i^{p_\phi} = \mathcal{F}^{-1}(\phi_{\tilde{x}_i} + p_{\phi_{y_i}}, \xi_{\tilde{x}_i})$, and $y_i$ is the ground-truth label of the adversarial example $\tilde{x}_i$. $\mathbb{I}(\cdot)$ denotes the indicator function, and $w_t$ is the weight during the t-th epoch. The Equation 1 is then incorporated with the designed weight as:

$$x^p = \mathcal{F}^{-1}(\phi_x + p_{\phi_y}, \xi_x + w_t p_{\xi_y}), \quad (7)$$

where we use Equation 7 for training. The finally learned weight $w^*$ is utilized for testing, and is shown in Table 5. The weighting strategy is illustrated in Figure 3.

---

**Algorithm 1** Phase and Amplitude-aware Prompting (PAP).

1: **Input:** The target model $h_\theta$, training dataset $\mathcal{D}$, batch size $n$, the number of batches $M$, epoch number $T$, perturbation budget $\epsilon$, the initialized phase-level prompts $\{p_{\phi_i}\}_{i=0}^{c-1}$ and amplitude-level prompts $\{p_{\xi_i}\}_{i=0}^{c-1}$.
2: **for** $t = 1$ **to** $T$ **do**
3:    **for** $m = 1$ **to** $M$ **do**
4:       Read mini-batch $\mathcal{B} = \{x_i\}_{i=1}^{n}$ from training set $\mathcal{D}$;
5:       Craft corresponding adversarial samples $\tilde{\mathcal{B}} = \{\tilde{x}_i\}_{i=1}^{n}$ at the given perturbation budget $\epsilon$;
6:       Calculate $L_{all}$ by Equation 8 to optimize $\{p_{\phi_i}\}_{i=0}^{c-1}$ and $\{p_{\xi_i}\}_{i=0}^{c-1}$;
7:    **end for**
8:    **if** $t \mod 5 = 0$ **then**
9:       update $w_t$ via Equation 6;
10:   **end if**
11: **end for**

---

### 3.3.3. OVERALL DEFENSE PROCEDURE

To improve the overall effectiveness of our combined defense, we incorporate the the weighting strategy into the training process. The overall loss function is denoted as:

$$\mathcal{L}_{all} = \mathcal{L}_{adv} + \lambda_1 \mathcal{L}_{nat} + \lambda_2 \mathcal{L}_{sim} + \lambda_3 \mathcal{L}_{mis}, \quad (8)$$

where $\lambda_1, \lambda_2, \lambda_3$ are hyper-parameters.

The overall defense procedure is presented in Algorithm 1. Specifically, during training, for each mini-batch $\mathcal{B}$, we craft adversarial examples $\tilde{\mathcal{B}}$. Then, we forward-pass $\mathcal{B}$ and $\tilde{\mathcal{B}}$ to calculate $L_{all}$ using Equation 8, and further optimize phase-level prompts $\{p_{\phi_i}\}_{i=0}^{c-1}$ and amplitude-level prompts $\{p_{\xi_i}\}_{i=0}^{c-1}$. The weight for amplitude-level prompts is adjusted by Equation 6. Through iteratively optimizing the prompts and adjusting the weights, the prompts are expected to provide superior robustness gains.

### 3.3.4. PROMPT SELECTION FOR TESTING

After acquiring our prompts, we need to explore an effective prompt selection method during testing. Previous methods (Chen et al., 2023) traverse all the prompts from all the classes for testing on naturally pre-trained models, which sets the label with the largest output among all the prompting cases as the final prediction (see Appendix B). However, it can easily cause high computational costs for testing on large datasets with numerous classes. To address it, we promote the test image to choose prompts corresponding to its predicted label directly. Incorporated with the learned weight, we obtain the prompted image for testing as:

$$x_{test}^p = \mathcal{F}^{-1}(\phi_{x_{test}} + p_{\phi_{y_{pred}}}, \xi_{x_{test}} + w^* p_{\xi_{y_{pred}}}), \quad (9)$$

where $p_{\phi_{y_{pred}}}$ and $p_{\xi_{y_{pred}}}$ are selected prompts from the predicted label $y_{pred}$. Since this strategy may result in mismatches between images and selected prompts, we introduce a data-prompt mismatching loss to alleviate its negative effects on robustness, which can be seen in Section 3.3.1.

*Table 3.* Robust accuracy (percentage) of defenses against adversarial attacks on CIFAR-10 and Tiny-ImageNet. The target models are ResNet18 and WRN28-10. We present the most successful defense results with **bold**.

| Defense | CIFAR-10 (ResNet18) | | | | Tiny-ImageNet (WRN28-10) | | | |
|---|---|---|---|---|---|---|---|---|
| | None | AA | C&W | DDN | None | AA | C&W | DDN |
| None | **94.83±0.05** | 0.00±0.00 | 0.00±0.00 | 0.00±0.00 | **66.62±0.11** | 0.00±0.00 | 0.00±0.00 | 0.02±0.00 |
| +Freq | 94.50±0.21 | 0.44±0.08 | 11.57±0.15 | 4.60±0.07 | 60.43±0.17 | 2.56±0.03 | 14.82±0.20 | 10.51±0.22 |
| +C-AVP | 92.67±0.51 | 0.61±0.11 | 1.93±0.07 | 1.05±0.17 | 66.52±0.02 | 0.39±0.00 | 5.83±0.07 | 4.19±0.25 |
| +PAP(Ours) | 87.12±0.21 | **37.34±0.11** | **80.27±0.28** | **66.22±0.21** | 57.30±0.15 | **5.33±0.07** | **42.14±0.36** | **33.27±0.41** |
| AT | 84.22±0.21 | 44.94±0.44 | 0.84±0.18 | 2.97±0.34 | 51.39±0.17 | 18.29±0.42 | 0.19±0.02 | 11.48±0.27 |
| +Freq | 78.26±0.11 | 51.50±0.31 | 35.67±0.35 | 35.12±0.17 | 44.84±0.28 | 22.93±0.26 | 19.25±0.41 | 21.76±0.20 |
| +C-AVP | 84.28±0.24 | 45.79±0.33 | 11.00±0.14 | 10.35±0.24 | 51.16±0.07 | 19.82±0.21 | 12.42±0.19 | 17.45±0.22 |
| +PAP(Ours) | **84.34±0.12** | **52.31±0.19** | **66.66±0.18** | **60.29±0.07** | **51.40±0.09** | **23.44±0.24** | **35.76±0.34** | **34.98±0.16** |
| TRADES | 81.59±0.21 | 48.98±0.23 | 0.74±0.10 | 5.03±0.09 | **48.98±0.07** | 17.87±0.07 | 0.15±0.00 | 14.29±0.09 |
| +Freq | 75.62±0.22 | 52.87±0.24 | 30.98±0.15 | 30.74±0.05 | 41.80±0.11 | 21.64±0.33 | 15.47±0.07 | 21.36±0.20 |
| +C-AVP | 81.58±0.17 | 49.44±0.24 | 4.47±0.20 | 7.97±0.35 | 48.59±0.19 | 19.15±0.34 | 9.18±0.17 | 18.57±0.13 |
| +PAP(Ours) | **81.62±0.14** | **54.36±0.24** | **63.89±0.27** | **57.58±0.20** | 48.03±0.24 | **22.73±0.34** | **30.43±0.14** | **31.81±0.26** |
| MART | **80.31±0.24** | 46.95±0.24 | 0.75±0.07 | 3.85±0.15 | **44.29±0.04** | 19.18±0.17 | 0.34±0.02 | 15.31±0.07 |
| +Freq | 74.14±0.18 | 52.60±0.22 | 34.40±0.16 | 32.79±0.19 | 37.76±0.28 | 22.80±0.27 | 15.46±0.24 | 21.75±0.24 |
| +C-AVP | 80.29±0.25 | 47.25±0.27 | 3.51±0.45 | 6.11±0.08 | 43.86±0.27 | 20.66±0.20 | 10.78±0.33 | 20.09±0.27 |
| +PAP(Ours) | 79.49±0.20 | **53.79±0.11** | **60.36±0.22** | **56.66±0.30** | 43.71±0.23 | **23.74±0.27** | **30.39±0.09** | **32.18±0.08** |

*Table 4.* Robust accuracy (percentage) of defenses against adversarial attacks on CIFAR-10 using the prompt selection method of C-AVP. The target model is ResNet18.

| Defense | None | AA | C&W | DDN |
|---|---|---|---|---|
| NAT | **94.83** | 0.00 | 0.00 | 0.00 |
| +Freq | 56.80 | 7.54 | 49.22 | 30.25 |
| +C-AVP | 52.17 | 32.26 | 46.78 | 39.51 |
| +PAP(Ours) | 86.59 | **38.33** | **83.88** | **70.24** |

*Table 5.* The learned weights for the amplitude-level prompts. We show the results of ResNet18 and WRN28-10.

| Model | Dataset | None | AT | TRADES | MART |
|---|---|---|---|---|---|
| ResNet18 | CIFAR-10 | 0 | 0.3054 | 0.2572 | 0.3258 |
| WRN28-10 | Tiny-ImageNet | 0 | 0.2702 | 0.3022 | 0.2848 |

## 4. Experiments

### 4.1. Experimental Settings

**Datasets and Models.** We use two popular benchmark datasets CIFAR-10 (Krizhevsky et al., 2009) and Tiny-ImageNet (Le & Yang, 2015) for defense evaluations. CIFAR-10 has 10 classes with 50,000 training images and 10,000 testing images, and Tiny-ImageNet has 200 classes with 100,000 training images, 10,000 validation images and 10,000 testing images. All the images are normalized into $[0, 1]$. We use ResNet18 (He et al., 2016) and WideResNet28-10 (WRN28-10) (Zagoruyko, 2016) as target models, and use WRN28-10, VGG19 (Simonyan & Zisserman, 2014) and popular Swin Transformer (Liu et al., 2021) for evaluating the transferability of defenses across different models.

**Attack Settings.** We use various adversarial attacks across two norms for evaluations. Specifically, we utilize $L_\infty$-norm AA (Croce & Hein, 2020), $L_2$-norm C&W (Carlini & Wagner, 2017) and $L_2$-norm DDN (Rony et al., 2019). The iteration number of $L_2$-norm DDN is set as 20, while that of $L_2$-norm C&W is 50. The perturbation budget for $L_\infty$-norm AA is set as $8/255$.

**Defense Settings.** We use prompt-based defenses C-AVP (Chen et al., 2023) and Freq (Huang et al., 2023) as baselines, where they are designed only for defending on naturally pre-trained models. We use the natural training (NAT), AT (Madry et al., 2018), TRADES (Zhang et al., 2019) and MART (Wang et al., 2019) to obtain pre-trained models. We use PGD with perturbation budget 8/255, perturb step 10 and step size 2/255 for training. We train prompts by SGD (Andrew & Gao, 2007) for 100 epochs, where the initial learning rate is 0.1 and is divided by 10 at the 75-th epoch. The batch size is 512 for CIFAR-10, and 256 for Tiny-ImageNet. We set $\lambda_1 = 3$, $\lambda_2 = 400$, $\lambda_3 = 4$ for naturally pre-trained models, and $\lambda_1 = 1$, $\lambda_2 = 5000$, $\lambda_3 = 4$ for adversarially pre-trained models. The threshold $\tau$ is set as 0.1, and we adjust the weights of amplitude-level prompts every 5 epochs. We omit deviations in several tables due to their small values ($\leq 0.60\%$). More details of settings can be found in Appendix C.

### 4.2. Defending against General Attacks

**Defending against White-box Attacks.** We apply various attacks to evaluate the robustness of our method and baselines. The average accuracies with the deviations are presented in Table 3.

Figure 4 shows our method can preserve complete semantic

*Table 6.* Robust accuracy (percentage) of defenses on different models. All the prompt-based defenses are trained on ResNet18, and then applied to the VGG19 and WRN28-10 respectively. We present the most successful defense results with **bold**.

| Model | Defense | CIFAR-10 | | | | Tiny-ImageNet | | | |
|---|---|---|---|---|---|---|---|---|---|
| | | None | AA | C&W | DDN | None | AA | C&W | DDN |
| VGG19 | NAT | **93.13** | 0.00 | 0.00 | 0.00 | **59.40** | 0.00 | 0.00 | 0.03 |
| | +Freq | 90.77 | 1.69 | 21.08 | 11.07 | 47.79 | 4.27 | 16.58 | 12.55 |
| | +C-AVP | 21.60 | 10.78 | 15.58 | 12.32 | 50.95 | 3.00 | 17.97 | 13.61 |
| | +PAP(Ours) | 87.65 | **34.28** | **81.33** | **65.65** | 49.30 | **7.39** | **38.66** | **31.26** |
| | AT | **80.12** | 42.61 | 0.38 | 3.42 | **38.99** | 10.73 | 0.13 | 5.40 |
| | +Freq | 73.78 | 49.04 | 34.08 | 33.10 | 30.11 | 15.69 | 16.23 | 16.93 |
| | +C-AVP | 79.99 | 43.60 | 10.67 | 10.41 | 38.86 | 14.20 | 16.85 | 16.60 |
| | +PAP(Ours) | 79.53 | **49.43** | **56.26** | **51.22** | 38.37 | **16.55** | **29.98** | **27.76** |
| WRN28-10 | NAT | **95.42** | 0.00 | 0.00 | 0.00 | **66.62** | 0.00 | 0.00 | 0.02 |
| | +Freq | 94.53 | 0.98 | 18.29 | 7.95 | 54.47 | 3.67 | 18.29 | 13.78 |
| | +C-AVP | 10.24 | 9.95 | 10.32 | 10.14 | 61.83 | 2.07 | 19.11 | 13.49 |
| | +PAP(Ours) | 87.83 | **41.81** | **81.88** | **69.19** | 54.93 | **6.88** | **40.90** | **34.05** |
| | AT | **87.85** | 49.45 | 1.16 | 4.62 | **51.50** | 18.27 | 0.19 | 11.48 |
| | +Freq | 82.50 | 54.40 | 34.79 | 34.54 | 44.29 | 23.32 | 19.88 | 22.18 |
| | +C-AVP | 87.74 | 50.50 | 12.01 | 12.93 | 51.18 | 20.03 | 15.31 | 19.31 |
| | +PAP(Ours) | 87.26 | **54.99** | **70.29** | **63.62** | 51.27 | **23.71** | **36.27** | **35.41** |

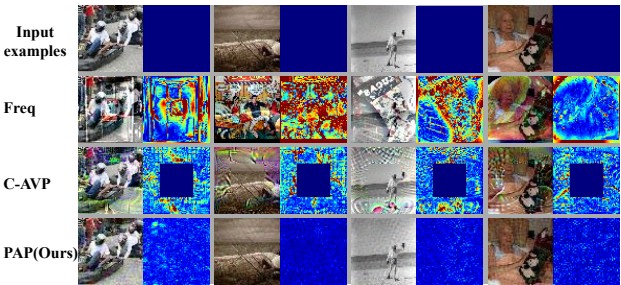

*Figure 4.* Visualizations of prompted images for input examples. For each pair of images, the left part denotes the prompted image, while the right part denotes the difference heatmap compared to the original input (*i.e.*, adversarial) example.

*Table 7.* Robust accuracy (percentage) of defenses against adaptive attacks on CIFAR-10. The target model is ResNet18.

| Defense | None | AdaA20 | AdaA40 |
|---|---|---|---|
| NAT+Freq | **94.35** | 0.68 | 0.88 |
| NAT+C-AVP | 77.74 | 0.01 | 0.01 |
| NAT+PAP(Ours) | 90.44 | **27.95** | **17.64** |
| AT+Freq | 78.30 | 32.44 | 32.22 |
| AT+C-AVP | 83.08 | 45.32 | 45.08 |
| AT+PAP(Ours) | **84.70** | **47.94** | **47.04** |

patterns after prompting. Quantitative analyses in Table 3 show that our method improves the robustness by a large margin on various attacks compared with existing defenses. On AutoAttack, our PAP helps increase the robust accuracy by about 37% on CIFAR-10 and 4% on Tiny-ImageNet for naturally pre-trained models, and achieves better robustness on adversarially pre-trained models from both datasets. Although Frequency Prompting improves robustness against AA for adversarially pre-trained models to some degrees, it decreases natural accuracy by about 7% on these models. On C&W and DDN, our method provides great positive effects for robustness. In addition, although our method sacrifices natural accuracy on naturally pre-trained models to some extent, it greatly improves robustness against all of these attacks (e.g., 80.27% and 42.14% against C&W on CIFAR-10 and Tiny-ImageNet).

**Defending against Black-box Attacks.** We apply transfer-

based attacks using VGG19 as the surrogate model and query-based attack Square (Andriushchenko et al., 2020) for evaluations. Table 12 in Appendix D shows our method achieves superior performances, verifying the practicality of our defense in real scenarios.

**Defenses on the Prompt Selection Method of C-AVP.** To further verify the stability of our method, we use the prompt selection strategy of C-AVP on the naturally pre-trained ResNet18 on CIFAR-10 for evaluations. Table 4 shows although previous methods achieve some improvements on robustness, they reduce natural accuracy by a large margin. In comparison, our method can still protect models more effectively without losing natural accuracy too much, achieving more stable performances.

### 4.3. Defense Transferability

To evaluate the transferability across different models, we applied our method trained on ResNet18 to other target models, *i.e.*, WRN28-10, VGG19 and Swin Transformer. Table 6 and Table 13 (see Appendix E) show our PAP can effectively help defend against various attacks across both convolutional neural networks and vision transformers. It in-

*Table 8.* The impact of different losses on CIFAR-10. We report average robust accuracies due to space limitations.

| Losses | | | NAT | | AT | |
|---|---|---|---|---|---|---|
| $\mathcal{L}_{nat}$ | $\mathcal{L}_{sim}$ | $\mathcal{L}_{mis}$ | None | Avg | None | Avg |
| × | √ | √ | 48.27 | 41.35 | 84.32 | **59.77** |
| √ | × | √ | 83.64 | **61.56** | 83.20 | 54.19 |
| √ | √ | × | 10.01 | 10.43 | 84.34 | 59.48 |
| √ | √ | √ | **87.12** | 61.27 | **84.34** | 59.57 |

*Table 9.* The effectiveness of the weight for prompting. We report average robust accuracies on both datasets.

| weight | CIFAR-10 | | Tiny-ImageNet | |
|---|---|---|---|---|
| | None | Avg | None | Avg |
| × | 75.92 | 57.27 | 41.59 | 28.70 |
| √ | **84.34** | **59.57** | **51.40** | **31.39** |

dicates that we can train our prompts only once and directly apply them to other models for defenses effectively.

### 4.4. Defending against Adaptive Attacks

Since we achieve defenses by prompting, the prompts could be leaked to attackers for performing adaptive attacks (AdaA). In this case, attackers focus on crafting adversarial noises for misleading predictions after prompting as:

$$\max_{\delta}\ell_{ce}(\mathcal{F}^{-1}(\mathcal{F}_{\phi}(x+\delta)+p_{\phi_y}, \mathcal{F}_{\xi}(x+\delta)+w_t p_{\xi_y}), y), \quad (10)$$

where $\ell_{ce}$ denotes the cross-entropy loss and $\delta$ is the perturbation. For fairness, we retrain our PAP on this attack and apply the same adaptive attack strategy on baselines using their own prompts to retrain them. Then, the retrained defenses are evaluated on adaptive attacks. The iteration number of attack for training is 10, while that for testing is 20 and 40. Table 7 shows our method achieves better robust accuracy, verifying the effectiveness of our method.

### 4.5. Ablation Studies

**Loss Functions.** We explore impacts of losses with different hyper-parameters. Table 8 shows removing any of these losses will damage performances, such as the extremely accuracy drop when removing $\mathcal{L}_{nat}$ or $\mathcal{L}_{mis}$ on NAT. Also, Appendix F shows natural and robust accuracies vary differently in various hyper-parameter settings for these losses. As a whole, the hyper-parameters we set achieve superior performances in both natural and robust accuracies.

**The Weighting Strategy.** To verify the effectiveness of the weighting strategy, we remove it for evaluations. As shown in Table 9, the performances drop a lot when removing the weighting strategy. Therefore, this strategy for dealing with different effects of phase-level and amplitude-level prompts on robustness is necessary and rational.

*Table 10.* The effectiveness of learning prompts for each class on CIFAR-10 using ResNet18 compared with **universal** prompts.

| Defense | None | AA | Defense | None | AA |
|---|---|---|---|---|---|
| NAT | **94.83** | 0.00 | AT | 84.22 | 44.94 |
| +Universal | 87.54 | 31.81 | +Universal | **84.56** | 51.92 |
| +PAP(Ours) | 87.12 | **37.34** | +PAP | 84.34 | **52.31** |

*Table 11.* The effectiveness of defenses with Gaussian Blur on CIFAR-10. The target model is ResNet18.

| Defense | None | AA | C&W | DDN |
|---|---|---|---|---|
| NAT | **70.57** | 25.92 | 56.79 | 40.10 |
| +Freq | 70.50 | 26.51 | 57.94 | 40.88 |
| +C-AVP | 65.83 | 25.75 | 53.01 | 37.04 |
| +PAP(Ours) | 67.83 | **46.57** | **65.38** | **58.85** |
| AT | **78.55** | 55.51 | 57.78 | 54.77 |
| +Freq | 72.72 | 55.94 | 57.57 | 55.61 |
| +C-AVP | 78.09 | 55.68 | 59.16 | 55.95 |
| +PAP(Ours) | 77.01 | **57.41** | **64.57** | **61.68** |

**Comparison with Universal Prompts.** To verify the superiority of learning prompts for each class, we train a universal phase-level prompt and a universal amplitude-level prompt for comparisons. For fairness, the data-prompt mismatching loss is removed on universal prompts, and other settings for universal prompts are the same as those from our method. Table 10 shows performances under universal prompts are worse than those of ours, indicating that our method helps enhance the robustness.

**Effectiveness When Blurring the Edges.** Some attacks like DDN tend to disrupt edges of objectives. Therefore, it's natural to question whether robustness gains from our PAP come from edge blurring. To this end, we apply Gaussian Blur on the test image for evaluations. As shown in Table 11, when blurring edges, our method can still achieve superior defenses, indicating the effectiveness of PAP does not come from edge blurring.

## 5. Limitation

Despite the advances in adversarial defenses, our method still has several limitations. First, our method sacrifices some natural accuracy when prompting on naturally pre-trained models. We will address it in the future such as using Contrastive Learning, since it is a useful method for mitigating the trade-off problem between natural and robust accuracies (Kim et al., 2020; Jiang et al., 2020; Xu et al., 2024). Second, we do not perform evaluations on ImageNet due to the limited computational resources. However, we conduct experiments on Tiny-ImageNet which has been widely used. Tiny-ImageNet-200 is larger and has a larger resolution than CIFAR-10, with more numbers of classes than those of ImageNet-100. Results show that our method

achieves superior performances, leading us to believe that our method can also work well on ImageNet. We leave them to the future work.

## 6. Conclusion

In this paper, we focus on specific semantic patterns for improving prompt-based defenses. It has been proven that phase and amplitude spectra reflect structures and textures, and both of them need to be manipulated for robustness. Therefore, we construct prompts using these spectra, and propose a Phase and Amplitude-aware Prompting (PAP) defense, which learns a phase-level prompt and an amplitude-level prompt for each class. Considering different influences of phase-level and amplitude-level prompts for robustness, we design a weighting method for them according to the robustness under these prompts. To perform testing efficiently, we select prompts according to predicted labels, and design a data-prompt mismatching loss to mitigate the negative effects of mismatches between images and their selected prompts. Experimental results demonstrate our method helps defend against general attacks and adaptive attacks, achieving superior transferability. Overall, our defense explores specific semantic patterns to improve performances of prompt-based defenses.

## Acknowledgements

This work was supported in part by the National Natural Science Foundation of China under Grants U22A2096, 62036007 and 62306227, in part by Scientific and Technological Innovation Teams in Shaanxi Province under grant 2025RS-CXTD-011, in part by the Shaanxi Province Core Technology Research and Development Project under grant 2024QY2-GJHX-11, in part by the Fundamental Research Funds for the Central Universities under Grant QTZX23042, in part by the Innovation Fund of Xidian University under Grant YJSJ25007.

## Impact Statement

This paper presents work whose goal is to advance the field of Machine Learning. There are many potential societal consequences of our work, none which we feel must be specifically highlighted here.

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

# A. Preliminary

**Notation.** We use capital letters like $X$ and $Y$ to represent random variables. Correspondingly, lower-case letters such as $x$ and $y$ are presented as the realizations of $X$ and $Y$. We use $\mathbb{B}(x, \epsilon)$ to denote the neighborhood of $x$: $\{\tilde{x} : \|x - \tilde{x}\| \le \epsilon\}$, where $\epsilon$ is the perturbation budget. Here, $\|\cdot\|$ represents the norm, which can be specified as $L_\infty$-norm $\|\cdot\|_\infty$ and $L_2$-norm $\|\cdot\|_2$. We define $f : \chi \to \{1, 2, ..., C\}$ as a classification function, where the $f$ can be parameterized by a deep neural network $h_\theta$ with the parameter $\theta$.

**Problem Setting.** In this paper, the task we focus on is the classification under adversarial settings, which means target models may be misled by adversarial noises. We sample natural data $\{(x_i, y_i)\}_{i=1}^n$ based on the distribution of $(X, Y)$, where $X$ and $Y$ are the variables of natural instances and their ground-truth labels. Here, $(X, Y) \in \chi \times \{1, 2, ..., c\}$ and $c$ is the number of classes. Given a deep neural network $h_\theta$ and a pair of natural data $(x, y)$, the adversarial example $\tilde{x}$ is crafted following such a constraint:

$$h_\theta(x) \ne y \quad s.t. \quad \|x - \tilde{x}\| \le \epsilon, \tag{11}$$

where $\tilde{x} = x + \delta$ and $\delta$ represents the adversarial noises. Since our focus is on attacking and defending for images, we utilize the Discrete Fourier Transform (DFT) and its inverse version (IDFT), denoted as $\mathcal{F}(\cdot)$ and $\mathcal{F}^{-1}(\cdot, \cdot)$, respectively. The phase and amplitude spectra are derived as $\phi_x = \mathcal{F}_\phi(x)$ and $\xi_x = \mathcal{F}_\xi(x)$. Specifically, we use $\phi_x$ and $\xi_x$ to denote the phase and amplitude spectra of a natural image $x$, while $\phi_{\tilde{x}}$ and $\xi_{\tilde{x}}$ represent the corresponding spectra of an adversarial example $\tilde{x}$. In addition, the process to recover an image from its phase and amplitude spectra is expressed as $x = \mathcal{F}^{-1}(\phi_x, \xi_x)$. Our goal is to design a set of prompts to assist the classification model $h_\theta$ in making accurate predictions. These prompts are trained without the need of model retraining, and are further utilized during testing.

# B. Prompt Selection Method for Testing from C-AVP

C-AVP (Chen et al., 2023) aims at utilizing pixel domains for prompting. It trains a prompt for each class, and designs a prompt selection method that traverses all the prompts from all the classes to get the final predictions for testing on naturally pre-trained models especially for CIFAR-10, which can be formulated as:

$$p = p_{i^*}, i^* = argmax_{i \in \mathcal{C}} h_\theta^i(x_{test} + p_i), \tag{12}$$

where $p$ is the selected prompt for the test image $x_{test}$, while $p_i$ is the prompt of class $i$ and $h_\theta^i$ is the output of class $i$, and $\mathcal{C}$ denotes the set of classes. Clearly, when the number of classes becomes large, this strategy for testing can easily cause extremely high computational costs. The prompt selection strategy of C-AVP is inefficient on numerous classes, and results in Table 4 show baselines with this strategy lose natural accuracy a lot. In comparison, our prompt selection strategy is efficient on numerous classes, and our defense with this strategy achieves superior defenses with higher natural accuracy, verifying the superiority of our prompt selection strategy.

# C. Experimental Settings

**Datasets and Models.** In this paper, we consider two popular benchmark datasets CIFAR-10 (Krizhevsky et al., 2009) and Tiny-ImageNet (Le & Yang, 2015). CIFAR-10 has 10 classes of images with a resolution of $32 \times 32$, which contains 50,000 training images and 10,000 testing images. The larger dataset Tiny-ImageNet has 200 classes with a resolution of $64 \times 64$ and has 100,000 training images, 10,000 validation images and 10,000 testing images. Images in all of these datasets are regarded as natural examples. We normalize all the images into the range of $[0, 1]$. Data augmentations including random crop and random horizontal flip are performed for all the data in the training stage. For the target model, we use ResNet18 (He et al., 2016) and WideResNet28-10 (WRN28-10) (Zagoruyko, 2016) for these datasets. We use WRN28-10, VGG19 (Simonyan & Zisserman, 2014) and a popular vision transformer architecture Swin Transformer (Liu et al., 2021) for evaluating the defense transferability across different models. The Swin Transformer is trained following previous studies about evaluating its defense performances (Liu et al., 2023).

**Attack Settings.** We introduce white-box attacks and black-box attacks to evaluate the defense. For white-box attack, we utilize $L_\infty$-norm AA (Croce & Hein, 2020), $L_2$-norm C&W (Carlini & Wagner, 2017) and $L_2$-norm DDN (Rony et al., 2019). The iteration number of $L_2$-norm DDN is set to 20, while that of $L_2$-norm C&W is 50. The perturbation budget for $L_\infty$-norm AA is $8/255$. For $L_\infty$-norm C&W, the learning rate is 0.01 and the confidence is 0. All the attacks mentioned above are set as non-targeted attacks. For black-box attacks, we apply transfer-based attacks under $L_\infty$-norm AA, $L_2$-norm

DDN using VGG19 as the surrogate model and query-based attacks under Square (Andriushchenko et al., 2020). The number of queries for Square is set to 200.

**Defense Settings.** We introduce two recently proposed prompt-based defenses C-AVP (Chen et al., 2023) and Freq (Huang et al., 2023) as baselines, which utilize the pixel domain and the frequency domain for prompting respectively. In addition, for the pre-trained models for optimizing and evaluating our prompts, we introduce natural training, AT (Madry et al., 2018), TRADES (Zhang et al., 2019) and MART (Wang et al., 2019). Note that all the pre-trained models are fixed without participating in any prompt training procedure. For the attack during training, we use PGD, where the perturbation budget and perturb step are 8/255 and 10, and the step size is 2/255. We train them using SGD (Andrew & Gao, 2007) for 100 epochs. The initial learning rate is 0.1 with batch size 512 for CIFAR-10, and batch size 256 for Tiny-ImageNet. The initial learning rate is divided by 10 at the 75-th epoch. We set $\lambda_1 = 3$, $\lambda_2 = 400$, $\lambda_3 = 4$ for naturally pre-trained models, and $\lambda_1 = 1$, $\lambda_2 = 5000$, $\lambda_3 = 4$ for adversarially pre-trained models.

## D. Defending against Black-box Attacks

We perform black-box attacks under transfer-based attacks and query-based attacks. The results are shown in Table 12. It is shown that our method achieve superior robust performances under black-box settings compared with baselines.

*Table 12.* Robust accuracy (percentage) of defenses against black-box attacks on CIFAR-10. The target model is ResNet18, and the surrogate model is VGG19. we perform AA and DDN as the transfer-based attack strategies.

| Defense | None | AA | DDN | Square |
|---|---|---|---|---|
| NAT | **94.83** | 16.91 | 53.46 | 22.10 |
| +Freq | 94.50 | 19.37 | 53.85 | 26.43 |
| +C-AVP | 92.67 | 17.27 | 52.87 | 22.64 |
| +PAP(Ours) | 87.12 | **51.90** | **74.79** | **61.38** |

## E. Defense Transferability to Vision Transformers

We further transfer our prompts trained on ResNet18 to popular Swin Transformer for evaluating the defense transferability of our method. As shown in Table 13, our method can be transferred well to vision transformers for improving their robustness, verifying the superior transferability across both convolutional neural networks and vision transformers.

*Table 13.* Robust accuracy (percentage) of our prompts transferred to vision transformers. The prompts are trained on ResNet18, and the vision transformer we introduced is Swin Transformer.

| Defense | None | AA | C&W | DDN |
|---|---|---|---|---|
| NAT | **88.98** | 0.00 | 0.00 | 0.00 |
| +Freq | 82.77 | 3.47 | 27.55 | 15.50 |
| +C-AVP | 30.33 | 9.00 | 17.28 | 12.90 |
| +PAP(Ours) | 84.82 | **10.13** | **77.97** | **49.15** |

## F. Hyper-parameter Studies

We perform several ablation studies for losses with different hyper-parameters as follows. For each hyper-parameter, it varies within a certain range while other hyper-paremeters are fixed. It can be seen that the natural and robust accuracies vary under different settings, and the hyper-parameters we set can achieve superior performances in both natural accuracy and robust accuracy.

There exists a trade-off problem in our method. As shown in Figure 5, for naturally pre-trained models, the natural accuracy increases while the robust accuracy drops as $\lambda_1$ or $\lambda_2$ increases. As shown in Figure 6, for adversarially pre-trained models, when $\lambda_2$ varies from 0 to 5000, the trade-off problem exists explicitly. Overall, the hyper-parameters we set achieve superior performances in both natural and robust accuracies.

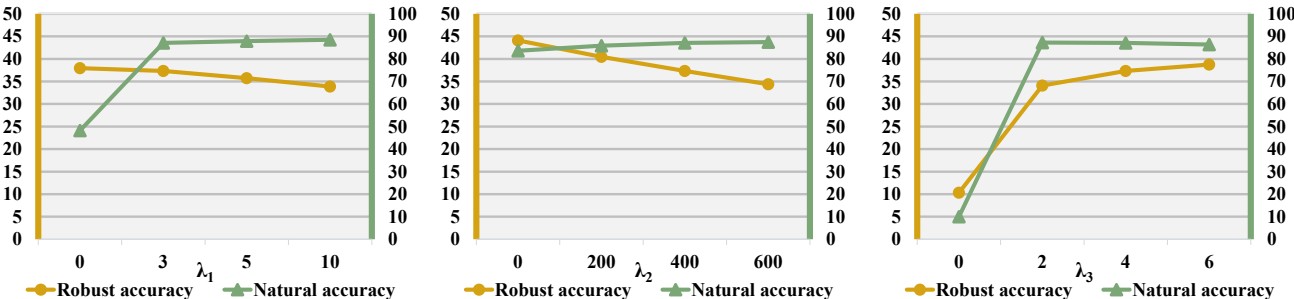

*Figure 5.* The impact of losses with different hyper-parameters on naturally pre-trained ResNet18 in CIFAR-10. For each hyper-parameter, it varies within a certain range while other hyper-parameters are fixed. We show the natural accuracy and robust accuracy against AA.

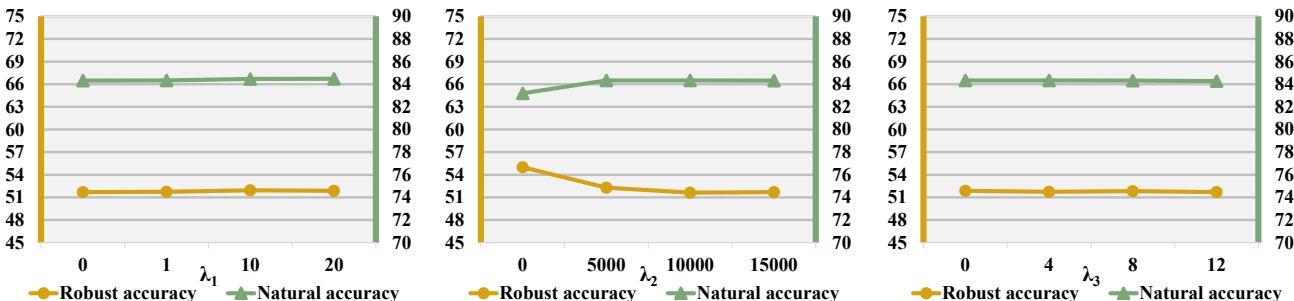

*Figure 6.* The impact of losses with different hyper-parameters on adversarially pre-trained ResNet18 in CIFAR-10. For each hyper-parameter, it varies within a certain range while other hyper-parameters are fixed. We show the natural accuracy and robust accuracy against AA.

## G. Visualizations of Prompted Images

We present additional visualized results of prompted images using our prompts, which are presented as follows. Here, following previous works (Elsayed et al., 2018; Tsai et al., 2020; Zhang et al., 2022), C-AVP performs prompting in the pixel space by adding random noises to the surrounding area inside the image, only keeping the square area in the center unchanged. Therefore, C-AVP is only a frame. It can be seen that our method retains complete and natural semantic patterns after prompting.

C-AVP performs prompting by adding noises around the image in the pixel domain, while Freq performs prompting on the high-frequency domain. They both train their prompts without considering their disruptions on the natural semantic patterns. In comparison, our method construct prompts on more specific semantic patterns, training them to enforce the prompted images to be as similar as possible to corresponding natural images. This can preserve more natural semantic patterns as shown in Figure 4, 7 and 8.

## H. Stability in Natural Accuracy

As a whole, our method performs more stably in natural accuracy. As shown in Section 4, baselines lose more natural accuracy under many cases, such as the worse transferability and performances under adaptive attacks of C-AVP and the natural accuray drop of Freq shown in Table 3 under adversarially pre-trained models. In comparison, our defense remains high natural accuracy in all of these cases, verifying the stability of our defense.

# I. Effectiveness on C&W

C&W method generates adversarial perturbations by performing optimizations in the pixel domain. Differently, our approach additionally considers the frequency domain. It disentangles the frequency domain information and leverages the amplitude and phase spectra as a way to focus more finely on important structural semantics and textures, which are not covered in the compared baselines. Therefore, our method can provide a more effective defense against perturbations generated by C&W.

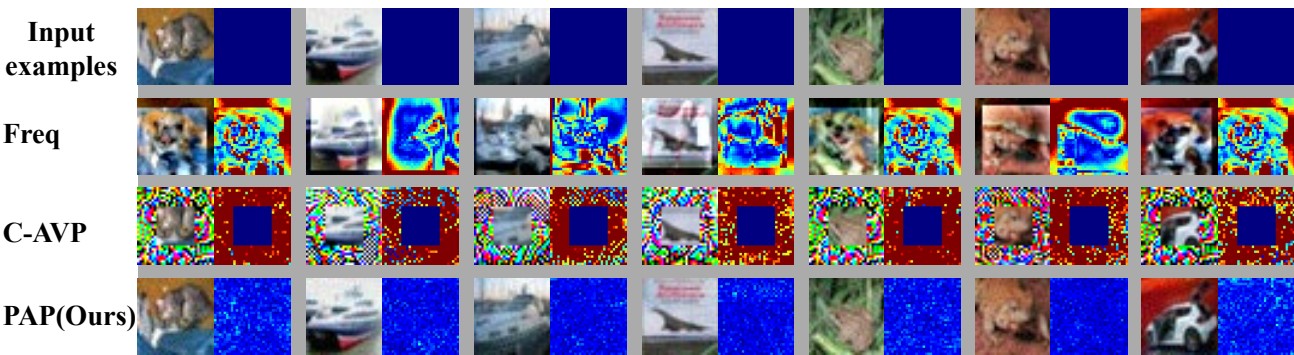

*Figure 7.* Visualizations of prompted images for input examples on CIFAR-10. The target model is naturally pre-trained ResNet18. For each pair of images, the left part denotes the prompted image, while the right part denotes the difference heatmap compared to the original input (*i.e.*, adversarial) example.

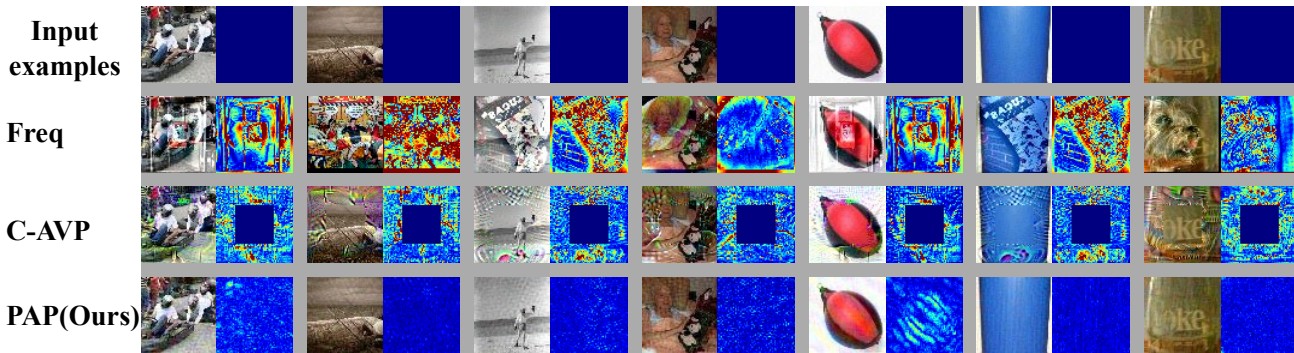

*Figure 8.* Visualizations of prompted images for input examples on Tiny-ImageNet. The target model is naturally pre-trained WRN28-10. For each pair of images, the left part denotes the prompted image, while the right part denotes the difference heatmap compared to the original input (*i.e.*, adversarial) example.

