# OpenReview forum: "Phase and Amplitude-aware Prompting for Enhancing Adversarial Robustness"
_ICML.cc/2025/Conference — ICML 2025 poster_

### Official Review · Reviewer_h61T · 2025-02-24

**Overall Recommendation:** 4

**Summary:**

This paper shows another defense based on visual prompting. They extended previous defenses applying the Fourier transform and integrating this in their defense design.
This method can outperform previous methods and showed its effectiveness against adaptive attacks.
The limitations could have been better discussed.

Nevertheless, in overall, I can recommend this paper.


## update after rebuttal
None

**Claims And Evidence:**

They extended a prompt defense with the Fourier transform and designed their own optimization algorithm.
In table 8, 9, 10, and 11, they showed empirically the effectiveness.

**Essential References Not Discussed:**

None.

**Experimental Designs Or Analyses:**

The classification models and datasets are sufficient, and the experimental design makes sense and also covering adaptive adversarial examples.

I do not understand, why C&W becomes so good in Table 6. When I look at Figure 2 in this paper: https://arxiv.org/pdf/2112.01601 then the FFT analysis does not seem so strong.  Any explanation?

**Methods And Evaluation Criteria:**

The proposed method has been adequately benchmarked. The scarification of the standard accuracy could be demonstrated more clearly.

**Other Comments Or Suggestions:**

ln12 noises -> Better perturbation
ln71 PGD -> better write PGD attack. PGD is just the optimization algorithm. And over time in related work - people use it wrongly.
Ln158 -> cite
- https://cse.buffalo.edu/~jsyuan/papers/2015/Learning%20LBP%20Structure%20by%20Maximizing%20the%20Conditional%20Mutual%20Information.pdf
- https://cims.nyu.edu/gcl/papers/ying2001tss.pdf

Is it necessary to give equation 2 and 3 so much space?
Ln325: L2 norm is not described the per budget?  0.5?

**Other Strengths And Weaknesses:**

Strength:
 - Effective method
 - Many experiments

Weaknesses:
 - The limitations could have been presented with greater transparency. In order to locate them, it was necessary to conduct a search, particularly with regard to the scarification of the standard accuracy. Additionally, the potential impact on larger datasets, such as ImageNet, could have been examined.
 -  The rationale behind the selection of adversarial attacks from a multitude of options remains opaque. The evaluation of PGD, while not an invariable practice, does occur on occasion. It is postulated that this practice will not engender a substantial disparity in the context of AA.

**Questions For Authors:**

1. Could you kindly explain your rationale for choosing PGD in table 2 over table 3?
2. Could you please explain why C-AVP is only a frame? It seems that on the CIFAR-10 dataset, the perturbation is in the middle of the image, and your FFT is applied globally.
3. Regarding limitations, it seems standard accuracy is being compromised. Could you please elaborate on the rationale behind suggesting contrastive learning in this context?
4. Could you please elaborate on the rationale behind the defense on C&W being so effective?

**Relation To Broader Scientific Literature:**

Prompting is a common technique for fine-tuning, particularly within the domain of NLP.

**Theoretical Claims:**

This paper is more closely aligned with the field of applied science, wherein an algorithm is identified and a loss function is formulated. Additionally, the Fourier transform and its inverse are applied to the entire image.
The request for additional theoretical proofs is not within the scope of the author's expertise.

---

> ### Author Rebuttal · Authors · 2025-03-30
>
> Thanks for your great efforts spending on reviewing our paper. Your comments are important to the improvement of our work, and we address them as follows:
>
> **The clarity on limitations.** We are sorry for the lack of clarity on limitations. The main limitation of our method is the sacrifice of natural accuracy. We would provide a clear introduction to this issue in a separate section. In addition, due to the limitation of computational resources, we do not perform evaluations on ImageNet and feel sorry for that. However, we conduct experiments on Tiny-ImageNet which has been widely used. Tiny-ImageNet-200 is larger and has a larger resolution than CIFAR-10, with more numbers of classes than those of ImageNet-100. Results show that our method achieves superior performances, leading us to believe that our method can also work well on ImageNet. We would state it in the updated version.
>
> **The selection of attacks.** Table 2 aims at verifying the different impacts of amplitude-level and phase-level prompts on robustness. In Table 2, we use PGD which is a popular method for evaluating the defense, to efficiently reflect the impacts of them on robustness. Table 3 and other evaluations in Section 4 aim at comprehensively evaluating the defense effectiveness of the proposed method. Therefore, we use the stronger AutoAttack which contains PGD to evaluate the proposed method comprehensively.
>
> **Typos.** We are sorry for these typos. We would correct them as follows:
> 1) Ln 12: *“Deep neural networks are found to be vulnerable to adversarial perturbations.”*
>
> 2) Ln 71: *“White-box attacks like Projected Gradient Descent (PGD) attack (Madry et al., 2018), AutoAttack (AA) (Croce & Hein, 2020), Carlini&Wagner (C&W) (Carlini & Wagner, 2017) and Decoupling Direction and Norm (DDN) (Rony et al., 2019) craft noises through accessing and utilizing models’ intrinsic information like structures and parameters.”*
>
> 3) Ln158: *“However, existing prompt-based defenses focus on mixed patterns like pixel or frequency information, which cannot capture specific patterns like structures and textures (Ying et al., 2001; Ren et al., 2015).”*
>
> 4) Equation 2 and 3: We are sorry for the waste of space here. We would move the "Classification Loss" section and these equations into the Appendix in the updated version.
>
> 5) Ln325: Yes, the default perturbation budget for L-2 norm is 0.5. We would supplement it in the updated version:
>
> *"The perturbation budget for L∞-norm AA is 8/255. And the perturbation budget for L-2 norm attacks is 0.5."*
>
> **Explanations of C-AVP.** Following previous works [1-4], C-AVP performs prompting in the pixel space by adding random noises to the surrounding area inside the image, only keeping the square area in the center unchanged. Therefore, C-AVP is only a frame. We would state it in the updated version.
>
> [1] Visual prompting: modifying pixel space to adapt pre-trained models. arXiv:2203.17274, 2022.
>
> [2] Adversarial reprogramming of neural networks. arXiv:1806.11146, 2018.
>
> [3] Transfer learning without knowing: reprogramming black-box machine learning models with scarce data and limited resources. ICML, 2020.
>
> [4] Fairness reprogramming. NeurIPS, 2022.
>
> **Rationale behind suggesting contrastive learning.** In the defense area, Contrastive Learning is a useful method for mitigating the trade-off problem between natural and robust accuracies, by learning the invariant natural semantic information between natural and adversarial examples through contrastive losses [5-7]. Therefore, we suggest Contrastive Learning and hope it can help break the limitation of our method about sacrificing natural accuracy when performing defenses. We would state it in the updated version.
>
> [5] Adversarial self-supervised contrastive learning. NeurIPS, 2020.
>
> [6] Robust pre-training by adversarial contrastive learning. NeurIPS, 2020.
>
> [7] Enhancing adversarial contrastive learning via adversarial invariant regularization. NeurIPS, 2024.
>
> **Effectiveness on C\&W.** C\&W method generates adversarial perturbations by performing optimizations in the pixel domain.
>  Differently, our approach additionally considers the frequency domain.  It disentangles the frequency domain information and leverages the amplitude and phase spectra as a way to focus more finely on important structural semantics and textures, which are not covered in the compared baselines. Therefore, our method can provide a more effective defense against perturbations generated by C\&W. We would supplement this statement in the updated version.
>
> We sincerely hope our answers can solve your concerns and obtain your increase in the score.

---

> > ### Comment · Reviewer_h61T · 2025-04-04
> >
> > Thanks for answering my questions in detail.
> >
> > I agree partly with this statement: "stronger AutoAttack which contains PGD". What do you mean by saying strong?
> > AutoAttack, especially the PGD variant APGD in AutoAttack is designed to have a reliable evaluation.

---

> > > ### Author Response · Authors · 2025-04-07
> > >
> > > Thank you very much for reviewing our responses. We are sorry for the inaccurate representation “strong”. AutoAttack, which contains APGD, FAB and Square attacks, is designed to perform a reliable evaluation. What we mean to express is that we use the AutoAttack to evaluate the effectiveness of the proposed method reliably in Section 4. Thank you for highlighting this representation!

---

### Official Review · Reviewer_9KjN · 2025-03-03

**Overall Recommendation:** 4

**Summary:**

This work exploits a prompt-based defense using specific texture and structure patterns, and proposes to incorporate these prompts with appropriate prompting weights according to their effects on robustness, which enhances the robustness in various scenarios with superior transferability across various networks.

**Claims And Evidence:**

Yes, the claims made in the submission are supported by clear and convincing evidence.

**Essential References Not Discussed:**

No, there are not related works that are essential but are not currently cited or discussed.

**Experimental Designs Or Analyses:**

Yes, I checked the validity of these designs and analyses, and consider that they are all sound.

**Methods And Evaluation Criteria:**

Yes, the proposed method and evaluation criteria all make sense for problems and applications.

**Other Comments Or Suggestions:**

(1) In the caption of Table 1, the word “indicate” after “Nat. Pha./Amp.” should be “indicates”.
(2) In the 4-th paragraph of Introduction, it mentions “phase and amplitude prompts”, which is mentioned as “phase and amplitude-level prompts” in the context.

**Other Strengths And Weaknesses:**

Strengths
+ Novelty: Introducing the popular field of prompts provides a novel and feasible idea for defense efficiently compared with adversarial training and denoising methods.
+ Incorporations of different semantic patterns: Incorporating prompts from different semantic patterns based on their influences on robustness exploits the benefits of these patterns innovatively.
+ Transferability: The proposed defense achieves superior transferability across convolutional neural networks and vision transformers compared with previous prompt-based defenses, verifying its practicality.

Weaknesses
- Few marginal results: In the experiments, while the proposed defense improves the robustness against various attacks a lot, it seems that the proposed method does not explicitly outperform the baseline named “Freq” in a few scenarios.
- Explanations about the superiority of the prompt selection strategy: The drawback of the prompt selection strategy of C-AVP for testing needs to be clarified comprehensively. It is introduced that this strategy is inefficient on numerous classes, and baselines using it sacrifice natural accuracy by a large margin. Bringing more clarity here can help further verify the superiority of the proposed method.

**Questions For Authors:**

(1) How does the Freq and C-AVP train for comparing with our method?
(2) It has been shown that the proposed method for learning prompts for each class achieve better results against AA compared with universal prompts. Does this superiority still exist against other attacks?

**Relation To Broader Scientific Literature:**

The key contributions of this paper are related to the findings of the semantic meanings about phase and amplitude spectra, and they are utilized as prompting, which is more innovative and efficient compared with previous adversarial training and denoising methods.

**Theoretical Claims:**

Yes, I checked the correctness of them, and they are all reasonable and correct without any problem.

---

> ### Author Rebuttal · Authors · 2025-03-30
>
> Thanks for your valuable comments. The responses to your concerns are as follows:
>
> **The superiority of our method.** Our method indeed does not outperform “Freq” by a large margin in a few scenarios. However, as shown in the Table 3, 4, 6, 7, 12 and 13, Freq sacrifices natural accuracy by a large margin in many cases. In comparison, the natural accuracy drop of our method is fewer, and our method achieves superior defenses in almost all of the defense cases.
>
> **The superiority of the prompt selection strategy.** The prompt selection method of C-AVP will become extremely inefficient on numerous classes. Meanwhile, as shown in Table 4, baselines using this strategy sacrifice natural accuracy by a large margin, further verifying its limitations on the model’s performances. Instead, our prompt selection strategy is efficient on numerous classes, and our defense using this strategy achieves superior defense without losing natural accuracy too much. We would state it in the updated version:
>
> *"The prompt selection strategy of C-AVP is inefficient on numerous classes, and results in Table 4 show baselines with this strategy lose natural accuracy a lot. In comparison, our prompt selection strategy is efficient on numerous classes, and our defense with this strategy achieves superior defenses with higher natural accuracy, verifying the superiority of our prompt selection strategy."*
>
> **Minor mistakes.** We are sorry for these mistakes. We would correct them:
> 1) *“Nat. Pha./Amp. indicates we replace phase/amplitude spectra with corresponding natural spectra.”*
> 2) *“Motivated by the above studies, we propose a Phase and Amplitude-aware Prompting (PAP) defense mechanism, which constructs phase and amplitude-level prompts to stabilize the model’s predictions during testing.”*
>
> **Training strategy of baselines.** For baselines Freq and C-AVP, we train them following the settings from their original papers without any modification for a fair comparison.
>
> **The superiority compared with universal prompts.** The defense superiority of our method still exists against attacks which are different from AA. As shown in the table below, our method achieves superior defenses against various attacks compared with universal prompts.
> |  | None | L∞-norm PGD | L2-norm PGD |
> |---|:---:|:---:|:---:|
> | NAT | **94.83** | 0 | 0.37 |
> |   +Universal | 87.54 | 30.23 | 57.05 |
> |   +PAP(Ours) | 87.12 | **35.45** | **58.94** |
>
> We sincerely hope our answers can solve your concerns and obtain your increase in the score.

---

### Official Review · Reviewer_fSJb · 2025-03-03

**Overall Recommendation:** 4

**Summary:**

This work proposes a prompting method for defense, through training prompts for each class using specific semantic patterns including structures and textures based on the Fourier Transform, which successfully defends against various general and adaptive attacks.

## Update After Rebuttal

The authors have addressed all my concerns comprehensively in their rebuttal. In particular, the authors clarify the stability of natural accuracy under different scenarios (e.g., transferability, adaptive attacks) with additional comparisons to baselines and reorganized hyperparameter settings for better readability. After reviewing the rebuttal, I maintain this score.

**Claims And Evidence:**

Yes, these claims are supported by clear and convincing evidence.

**Essential References Not Discussed:**

No. There aren’t related works that are essential but are not discussed in the paper.

**Experimental Designs Or Analyses:**

Yes, I checked the soundness of all the experimental designs or analyses and concluded that they are sound and valid.

**Methods And Evaluation Criteria:**

Yes, the proposed method and criteria make sense for the problem or application.

**Other Comments Or Suggestions:**

A few typos about the consistency of phrases need to be considered, i.e., “tested image” in the ablation study which is “test image” in the context.

**Other Strengths And Weaknesses:**

**Strengths:**

1. The proposed defense strategy firstly exploits visual prompts from specific phase and amplitude spectra, which explores a promising direction for efficient defenses utilizing specific semantic patterns.
2. Experiments from various viewpoints including adaptive attacks and transferability evaluation verify the effectiveness of the proposed method in defenses.
3. The training and testing procedures are introduced with a reasonable and sound logic, where they are analyzed and proven to be effective by sufficient defense evaluations and ablation studies.

**Weaknesses:**

1. The stability of the proposed method in natural accuracy under various scenarios needs to be cleared. It is clear that the method provides great robust gains while reducing a few natural performances which is mentioned as limitations. In fact, baselines lose more natural accuracy when evaluating on transferability and adaptive attacks, which needs to be discussed.
2. The threshold, hyper-parameters and the frequency of adjusting weights are presented in a somewhat dispersed manner. Maybe they can be centered in experimental settings so that readers can notice these settings efficiently.

**Questions For Authors:**

1. What are the experimental settings for adaptive attacks when training and testing, except for the number of iterations?
2. I wonder if the designed data-prompt mismatching loss contradicts the classification loss considering the different assignments of prompting of different losses for training.
3. What are the settings of gaussian blur performed in the ablation studies?

**Relation To Broader Scientific Literature:**

The key contributions of prompting on phase and amplitude spectra are related to the previous findings that analyzed the phase and amplitude spectra of images hold structures and textures respectively.

**Theoretical Claims:**

Yes, I checked the correctness of them and found that there are no issues among them.

---

> ### Author Rebuttal · Authors · 2025-03-30
>
> Thanks for your constructive suggestions. Your comments are important to our work, and we address them as follows:
>
> **Stability in natural accuracy.** Our method lose a few natural accuracy when performing defenses. However, as shown in the Section 4, baselines lose more natural accuracy under various scenarios, such as the performances of C-AVP under transferability evaluations and those of Freq in Table 3. In comparison, our method remains high natural accuracy in these scenarios, achieving more stable performances in natural accuracy. We would state it in the updated version:
>
> *"As a whole, our method performs more stably in natural accuracy. As shown in Section 4, baselines lose more natural accuracy under many cases, such as the worse transferability and performances under adaptive attacks of C-AVP and the natural accuray drop of Freq shown in Table 3 under adversarially pre-trained models. In comparison, our defense remains high natural accuracy in all of these cases, verifying the stability of our defense."*
>
> **The introduction of experimental settings.** We are sorry for such a presentation that is not easy to read. We would introduce all of them in the section of experimental settings clearly in the updated version:
>
> *"We set λ1=3, λ2=400, λ3=4 for naturally pre-trained models, and λ1=1, λ2=5000, λ3=4 for adversarially pre-trained models. The threshold τ is set as 0.1, and we adjust the weights of amplitude-level prompts every 5 epochs."*
>
> **The consistency of the phrase.** We are sorry for the inconsistency. We would correct the phrase:
>
> *“To this end, we apply Gaussian Blur on the test image for evaluations.”*
>
> **Settings of adaptive attacks.** During training and testing, for other settings of adaptive attacks, the perturbation budget is 8/255, and the step size is 2/255.
>
> **Influences of the data-prompt mismatching loss.** The data-prompt mismatching loss does not contradict the classification loss. As shown in Figure 5, when the λ3 varies from 0 to 4, both the natural and robust accuracies increase by a large margin. Therefore, training prompts using these losses simultaneously can achieve superior performances in both natural and robust accuracies.
>
> **The settings of Gaussian Blur in the ablation study.** For the Gaussian Blur performed in the ablation study, the kernel size is set as 3, and the standard deviation is sampled randomly from 0.1 to 2.0. The Gaussian Blur is conducted on the test image for evaluations.
>
> We sincerely hope our answers can solve your concerns and obtain your increase in the score.

---

### Official Review · Reviewer_pd1Q · 2025-03-08

**Overall Recommendation:** 4

**Summary:**

This paper proposes a defense strategy based on prompting on structures and textures, with appropriate weights adjusted by their influences on robustness for incorporating their benefit for defenses. It achieves superior defense performances on general and adaptive attacks and defense transferability.

**Claims And Evidence:**

Yes, the claims made are all supported by clear evidence.

**Essential References Not Discussed:**

No, there are not related works that are essential but are not discussed

**Experimental Designs Or Analyses:**

Yes, I checked the validity of these designs and consider they are reasonable and sound.

**Methods And Evaluation Criteria:**

Yes, the proposed method with the criteria makes sense for the problem and application.

**Other Comments Or Suggestions:**

A minor grammar mistake exists in Appendix B, where the phrase “the number of class” may need to be modified as “the number of classes”.

**Other Strengths And Weaknesses:**

Strengths
+ The framework and its corresponding motivations are clearly presented, with precise expressions of formulas and figures.

+ Given the inefficiency of traditional defenses, the area the authors focus on is novel, where the use of prompting on specific semantics with appropriate weights offers promising directions for exploration.

+ The experiments presented in the manuscript comprehensively validate the effectiveness of the method from multiple perspectives.

Weaknesses
- Authors should discuss the potential problem of trade-off between natural accuracy and robustness under the proposed method. The proposed defense improves robustness by a large margin according to the presented results, while the trade-off exists according to hyper-parameter studies, which deserves to be discussed briefly for the integrity of logic.

- The analyses of visualizations of prompted images may need to be cleared further. The baselines perform prompting in a way that is different from the proposed defense, and analyzing their limitations in preserving natural semantic patterns is somewhat necessary.

- The fairness of the ablation study comparing with universal prompts needs to be addressed. When training universal prompts, the data-prompt mismatching loss cannot be applied, and this may need to be removed for a fair comparison. Please clarify this point.

**Questions For Authors:**

- In the experiments, I am concerned about the attack settings of adaptive attacks, considering the fairness of evaluations.
- Directly applying prompts to the amplitude and phase spectra is a novel approach. However, could this method potentially disrupt the integrity of these spectra?

**Relation To Broader Scientific Literature:**

The key contributions of the paper are related to previous analyses about the semantic patterns of phase and amplitude spectra. They are utilized in the submission for mitigating the negative effects on specific semantic patterns to enhance the robustness of prompt-based defenses.

**Theoretical Claims:**

Yes, I checked the correctness and found that they are all correct.

---

> ### Author Rebuttal · Authors · 2025-03-30
>
> Thanks for your valuable comments and constructive suggestions. The responses to your concerns are as follows:
>
> **Discussions on the trade-off problem.** Our method has a trade-off problem under different hyper-parameters. On naturally pre-trained models, when λ1 increases, the natural accuracy increases while the robust accuracy drops explicitly. As λ2 increases, the robust accuracy drops a lot while the natural accuracy does not change too much for the naturally pre-trained models. On adversarially pre-trained models, the trade-off problem only exists when λ2 varies. Overall, the selected hyper-parameters achieve superior performances in both natural and robust accuracies. We would state it in the updated version:
>
> *"There exists a trade-off problem in our method. As shown in Figure 5, for naturally pre-trained models, the natural accuracy increases while the robust accuracy drops as λ1 or λ2 increases. As shown in Figure 6, for adversarially pre-trained models, when λ2 varies from 0 to 5000, the trade-off problem exists explicitly. Overall, the hyper-parameters we set achieve superior performances in both natural and robust accuracies."*
>
> **Analyses of visualizations of prompted images.** C-AVP performs prompting only by adding noises around the image in the pixel space to mitigate the negative effects of adversarial perturbations. Frequency Prompting method directly adds frequency prompts on the high-frequency domain. They both construct and train their prompts without considering their disruptions on the natural semantic patterns in the pixel space. In comparison, our method constructs prompts on the more specific semantic patterns (i.e., textures from amplitude spectra and structures from phase spectra), and trains them using a loss which enforces the prompted images to be as similar as possible to the corresponding natural images in the pixel space. To this end, our method preserves more natural semantic patterns after prompting compared with baselines. We would state it in the updated version:
>
> *"C-AVP performs prompting by adding noises around the image in the pixel domain, while Freq performs prompting on the high-frequency domain. They both train their prompts without considering their disruptions on the natural semantic patterns. In comparison, our method construct prompts on more specific semantic patterns, training them to enforce the prompted images to be as similar as possible to corresponding natural images. This can preserve more natural semantic patterns as shown in Figure 4, 7 and 8."*
>
> **Fairness of the comparison with universal prompts.** The data-prompt mismatching loss indeed cannot be performed on universal prompts. For fairness, we remove the data-prompt mismatching loss on universal prompts, and other training settings for universal prompts are the same as those from our method. We would state it in the updated version:
>
> *"For fairness, the data-prompt mismatching loss is removed on universal prompts, and other settings for universal prompts are the same as those from our method."*
>
> **Typos.** We are sorry for this typo. We would correct the typo:
>
> *“Clearly, when the number of classes becomes large, this strategy for testing can easily cause extremely high computational costs.”*
>
> **Settings of adaptive attacks.** For adaptive attacks on training and testing, the perturbation budget is 8/255 and the step size is 2/255. The iteration number is 10 for training, and 20 and 40 for testing.
>
> **The integrity of spectra after prompting.** We conduct a reconstruction loss to train the prompts, enforcing the prompted images to be as similar as possible to the corresponding natural images in the pixel space. Therefore, our method does not disrupt the integrity of these spectra, which can be shown in the Figure 4, 7 and 8.
>
> We sincerely hope our answers can solve your concerns and obtain your increase in the score.

---

### Decision · Program_Chairs · 2025-05-01

**Decision:**

Accept (poster)

**Comment:**

This paper proposes PAP (Phase and Amplitude-aware Prompting), a novel prompting-based defense that enhances adversarial robustness by leveraging class-specific prompts constructed from phase (structure) and amplitude (texture) spectra. Prompt weights are adaptively optimized during training based on robustness, and selected at inference using predicted labels. The method achieves strong performance across various attacks and datasets, demonstrating improved robustness and transferability with greater efficiency and interpretability.

The reviewers unanimously recommended acceptance and praised the paper's originality, strong empirical results, and well-structured methodology. Key strengths include the effective use of semantically meaningful prompts, solid evaluations against general and adaptive attacks, and fair comparisons to baselines. Minor concerns were raised about clarity regarding natural accuracy trade-offs, adaptive attack configurations, and the organization of hyperparameter details. These were adequately addressed in the rebuttal.